# The Role of Monoclonal Antibodies in Smoldering and Newly Diagnosed Transplant-Eligible Multiple Myeloma

**DOI:** 10.3390/ph13120451

**Published:** 2020-12-10

**Authors:** Elena Zamagni, Paola Tacchetti, Paola Deias, Francesca Patriarca

**Affiliations:** 1Istituto di Ematologia “Seragnoli”, IRCCS, Azienda Ospedaliero-Universitaria di Bologna, 40138 Bologna, Italy; paola.tacchetti2@unibo.it; 2Clinica Ematologica e Unità di Terapie Cellulari, Azienda Sanitaria Universitaria Friuli Centrale, 33100 Udine, Italy; paola.deias@asufc.sanita.fvg.it (D.P.); francesca.patriarca@asufc.sanita.fvg.it (P.F.); 3Dipartimento di Area Medica, Università di Udine, 33100 Udine, Italy

**Keywords:** multiple myeloma, monoclonal antibodies, minimal residual disease, survival outcomes

## Abstract

The recent introduction of monoclonal antibodies (MoAbs), with several cellular targets, such as CD-38 (daratumumab and isatuximab) and SLAM F7 (elotuzumab), differently combined with other classes of agents, has significantly extended the outcomes of patients with multiple myeloma (MM) in different phases of the disease. Initially used in advanced/refractory patients, different MoAbs combination have been introduced in the treatment of newly diagnosed transplant eligible patients (NDTEMM), showing a significant improvement in the depth of the response and in survival outcomes, without a significant price in terms of toxicity. In smoldering MM, MoAbs have been applied, either alone or in combination with other drugs, with the goal of delaying the progression to active MM and restoring the immune system. In this review, we will focus on the main results achieved so far and on the main on-going trials using MoAbs in SMM and NDTEMM.

## 1. Introduction

The treatment of multiple myeloma (MM) has been characterized in the last 20 years by the successive introduction of novel classes of agents, as part of the therapy of both relapsed/refractory (RRMM) and newly diagnosed (NDMM) patients [1]. In particular, the recent introduction of monoclonal antibodies (MoAbs), with several cellular targets, such as daratumumab (Dara) and elotuzumab (Elo), differently combined with other classes of agents, has significantly extended the outcomes of patients in different phases of the disease [2,3].

Smoldering MM (SMM) is an asymptomatic precursor to active MM [4]; the standard of care for patients with SMM has traditionally been observation until a myeloma-defining event (MDE) occurs [5]. However, since two trials demonstrated a significant improvement in the survival outcomes of high-risk SMM with the early use of the immunomodulatory agent (IMiD) lenalidomide, either alone or in combination with dexamethasone [6,7], several phase II and III trials have been planned, with the goal either of eradicating the disease (“curative” approach) or of delaying the progression to active MM and restoring the immune system [8]. The data available up to now on the use of MoAbs belong to this second approach.

The choice of first-line therapy in ND transplant eligible MM (NDTEMM) patients is of a priority importance, if one is to achieve the longest disease control and the best quality of life [9]. In this regard, the addition of MoAbs to the standard triplets has in several prospective trials proved to significantly deepen the level of response, in particular of the minimal residual disease (MRD) negativity rate, and ultimately extend progression-free survival (PFS) [10]. Moreover, given the mechanism of action and safety profile of MoAbs, the shift from triplets to quadruplets has not resulted in a worsening of patients’ quality of life [2,3]. In this review, we will focus on the main results achieved so far and on the main on-going trials using MoAbs in SMM and NDTEMM.

## 2. Discussion 

### 2.1. Monoclonal Antibodies in Smoldering Multiple Myeloma

The bone marrow (BM) microenvironment of patients with MM is usually characterized by diminished immune cell function [11]; consequently, activating the immune system to target myeloma cells via immunotherapy seems an intriguing approach. Dara and Elo have been tested in phase 2 and 3 clinical trial in this subset of patients, with the goal of delaying progression to active MM [12]. The results of the main studies with Dara and Elo are reported in Table 1 and Table 2, respectively.

### 2.2. Anti-CD-38 Monoclonal Antibodies

A single study tested the efficacy of Dara in delaying progression from high and moderate risk SMM to MM [13]. The CENTAURUS study (NCT02316106) consisted of a phase 2 randomized study employing 3 different Dara schedules. One hundred twenty three patients, selected on the basis of the Mayo Clinic risk stratification of SMM and the absence of CRAB and SlimCRAB signs, received 8-week cycles of Dara 16 mg/kg intravenously following an intense (20 cycles, 32 Dara administrations) or intermediate (20 cycles, 26 Dara administrations) or short schedule (1 cycle, 8 Dara administrations). Two co-primary endpoints were chosen: rate of complete remission (CR)s and rate of progressive disease (PD)s plus death at 6 and 12 months after randomization of the last patient, respectively. The first end-point was not met, since CRs were far less than the established cut-off of 15% in any arm of the study (2.4%, 4.9%, and 0% for intense, intermediate, and short dosing), suggesting that Dara monotherapy was not sufficient to eradicate SMM. These results were much inferior to those attained by treatment with carfilzomib, lenalidomide and dexamethasone (KRd), which achieved up to 60% CR and MRD negativity [14,15]. The 24-month PFS rates were 89.9%, 82.0%, and 75.3%, respectively, indicating that the prolonged schedule delayed clinical and biochemical progression more efficiently than the short schedule. The safety profile was one already known in RRMM, leading to protocol interruption in less than 10% of patients in any arm. The most common severe adverse events were hypertension and hyperglycemia. Dara monotherapy proved to control progression of SMM without achieving eradication of the plasma cell clone. These results were similar to the QuiRedex protocol, which produced only 4% very good partial response (VGPR), but significantly reduced progression in comparison with active monitoring [16].

On the basis of the clinical results of this phase 2 study, a phase 3 AQUILA study (NCT03301220) has been designed: it will test the subcutaneous administration of Dara in comparison with observation in SMM selected on the basis of the most recent Mayo Criteria [17].

A combination phase 2 trial of Dara with KRd (ASCENT trial, NCT03289299) is currently on-going for patients with high-risk SMM, the primary end-point being rate of stringent CR (sCR).

### 2.3. Anti-SLAM F7 Monoclonal Antibodies

Elo, the main anti SLAM F7 MoAb, failed to show any substantial efficacy as a single agent in the treatment of advanced/refractory MM and is recommended in combination with lenalidomide and dexamethasone (len-dex) for RR patients after 1–3 lines of therapy [18]. As per its mechanism of action, stimulating natural killer (NK) cells and augmenting antibody-dependent cell-mediated cytotoxicity (ADCC) [19], it has been speculated that Elo might be more effective earlier in the course of MM, when immune function is more preserved. CD56dim/CD16+/CD3-/CD45+ NK cells represent the primary subset responsible for Elo-induced ADCC.

A phase 2 study [20] (NCT02960555) investigated the efficacy of Elo monotherapy at 2 different doses (20 mg/kg, days 1-8 cycle 1, then monthly or 10 mg/kg, weekly cycles 1 and 2 and every 2 weeks thereafter) in 31 patients with high-risk SMM, according to the 2010 International Myeloma Working Group (IMWG) criteria [21]. The primary endpoint was the relationship between the proportion of BM-derived CD56dim NK cells and maximal M protein reduction; secondary endpoints included overall response rate (ORR) and PFS. No significant relationship emerged between CD56dim NK cell proportion and M protein decrease. With a minimum follow-up of 28 months, the ORR was 10%, while the 2-year PFS rate was 69%. It is difficult to interpret these results, especially considering that according to the updated risk stratification [4] 5/31 patients should have been considered as MM for the presence of more than 60% plasma cells while most of the remaining patients appeared to be predominantly “intermediate” risk, with a spontaneous projected median PFS of approximately 5 years [5].

Elo showed a synergistic activity with the combination of len-dex in RRMM [18], thus potentially being a better alternative for SMM. Elo-len-dex was tested in a phase 2 randomized study (NCT02279394) dedicated to high-risk SMM, according to the Mayo Clinic criteria or one of the new proposed criteria [5] (patients with one of the malignancy biomarkers were excluded). Fifty patients were enrolled and received Elo-len with or without the addition of dex [22]. The primary analysis of the trial, currently on-going, showed that 84% of the patients achieved at least a partial response (PR), in a median time of 2.8 months, with 6% CR and 37% VGPR. The response rate did not differ in accordance with the presence of cytogenetic high-risk features, defined according to the IMWG definition [4]. With a median follow-up of 29 months, the median PFS and overall survival (OS) were not attained and at 3 years none of the patients progressed to MM. The toxicity profile was manageable.

### 2.4. Monoclonal Antibodies in Newly Diagnosed Transplant-Eligible Multiple Myeloma

Although the majority of published results from combinations of MoAbs with other classes of agents refer to RRMM, several important trials have been run in NDMM eligible for autologous stem cell transplantation (ASCT) and many others are currently on-going or planned by the main cooperative groups. Dara in combination with bortezomib-thalidomide-dexamethasone (VTd) has been approved by both the US Food and Drug administration (FDA) and the European Medicine Agency (EMA) as induction therapy prior to ASCT. The combination of MoAb and a triplet (proteasome inhibitor (PI)-IMiD-dex) will probably become the new standard of care.

### 2.5. Anti-CD-38 Monoclonal Antibodies

#### 2.5.1. Daratumumab

The most mature clinical data about incorporation of a MoAb in an NDTEMM platform derive from a randomized multicenter open-label phase 3 trial (NCT02541383), comparing the Daratumumab plus VTd (Dara-VTd) quadruplet with the VTd triplet, a well -known “standard of care” in many countries [10]. The platform here consisted of 4 cycles of induction, mobilization after 3 g/m^2^ cyclophosphamide, with plerixafor allowed in case of inadequate collection, single ASCT and two cycles of consolidation. After consolidation, patients with PR or better underwent a second randomization between Dara maintenance every 8 weeks until progression or observation. The study population included 1080 patients, with a median age of 59 years in Dara-VTd group and 58 in VTd group and high-risk cytogenetics by FISH or ISS 3 in 15% of cases, respectively. The effectiveness was evaluated in terms of the proportion of patients achieving deep response (sCR and MRD negativity, tested by a multiparametric flow cytometry (MFC) approach with 10-5cut-off) after consolidation. The rate of sCR was significantly higher after Dara-VTd than with VTd (29% versus 20%; *p* = 0.001). The superiority was maintained across different subgroups, with the exception of high-risk (HR) patients, whether cytogenetic or International Staging System (ISS) based. Deep response improved over time, during the different treatment phases. The MRD negativity rate after consolidation, by MFC, was significantly higher in the Dara group, considering the whole study population (64% versus 44%, *p* < 0.0001), but also in the post-hoc analyses, evaluating only patients achieving ≥VGPR (62% versus 43%, *p* < 0.0001) or ≥ CR (34% versus 20%, *p* < 0.0001) [23]. Results were in favor of the Dara arm, even when MRD was evaluated with next generation sequencing (NGS) and a more stringent cut-off of 10^−6^ (57% versus 37, *p* < 0.0001). These deeper responses translated into a longer PFS, with the probability, at a median follow-up of 18 months, of 93% after Dara-VTd versus 85% after VTd (HR 0,47, 95% CI 0.33-0.67, *p* < 0.0001), confirmed in all pre-specified subgroups, including HR patients. In multivariate analyses, a PFS advantage was demonstrated in patients obtaining MRD negativity, regardless of the treatment group (HR 0.31, 95% CI 0.20–0.50, *p* < 0.0001). The data are still not mature for evaluating OS or the effect of maintenance. Tolerability was good in both arms, without significant differences in the rate of grade 3-4 adverse events or the treatment discontinuation rate.

A median of 6.3 × 106/kg CD34+ cells were collected after Dara-VTd in comparison with 8.9 × 10^6^/kg CD34+ cells after VTd. Although plerixafor was more frequently used in the Dara arm (22% versus 8%), there was no difference in the ASCT rate and the subsequent hematopoietic engraftment. Overall, the CASSIOPEIA study showed that the addition of Dara to standard of care induction and consolidation was effective, well tolerated and did not affect the possibility of performing ASCT, leading to FDA and EMA approval of Dara-VTd in NDTEMM.

CASSIOPET, a CASSIOPEIA companion study, evaluated concordance between two MRD detecting techniques (MFC and imaging by positron tomography/computed tomography, PET/CT by fluorodeoxyglucose, FDG) and compared PFS in patients reaching MRD negative status in the marrow or in both tests [24]. PET/CT was interpreted by a five-point Deauville score system, applied to bone lesions, bone marrow, extramedullary disease (EMD) and paramedullary disease (PMD), and maximum standard uptake value (SUV). PET/CT scans were available in 268 patients at baseline and in 184 patients after consolidation. At diagnosis, 20% of patients had a negative PET. The study confirmed that PET/CT negativity at baseline is associated with prolonged PFS (100% versus 92.5% at 12 months and 100% vs. 87.5% at 18 months). Double MRD negativity post consolidation was achieved in a greater proportion of patients in Dara arm than with VTd arm (66.7% versus 47.5%, *p* = 0.0105). The data are not mature enough to show a PFS advantage, but it can reasonably be expected, given the significantly higher proportion of patients achieving MRD negative status with Dara use in frontline treatment.

Dara has been tested in addition to other platforms of treatment for NDTEMM; however, the results available came from phase 1 or 2 trials and from the evaluation of smaller groups of patients.

The GRIFFIN study (NCT02874742), a phase 2 randomized trial, compared Dara-bortezomib-lenalidomide-dexamethasone (VRd) with VRd as induction (four cycles) and consolidation after ASCT, followed by maintenance with lenalidomide and Dara until disease progression or up to 2 years [25,26]. The safety run-in cohort of 16 patients showed an acceptable safety profile and was followed by the accrual of 207 patients. The primary end-point, sCR rate by the end of post-ASCT consolidation, favored Dara-VRd versus VRd (42.4% versus 32.0%; odds ratio 1.57; 95% CI 0.87–2.82; 1-sided *p* = 0.068) and met the prespecified 1-sided α of 0.10. With longer follow-up (median 22.1 months), responses deepened and sCR rates improved for Dara-VRd versus VRd (62.6% versus 45.4%; *p* = 0.0177). Moreover, the MRD-negativity (10^−5^ like before -5 as apex threshold) rate was higher in the Dara-VRd group than in the VRd group in the intent-to-treat population (51.0% versus 20.4%; *p* < 0.0001) as well as in pre-specified subgroups, even if the benefit was not statistically significant for HR patients. Median PFS was not reached, with 24-month PFS estimates of 95.8% (95% CI, 89.2–98.4) for the Dara-VRd group and 89.8% (95% CI, 77.1–95.7) for the VRd group. The positive results from GRIFFIN study support the ongoing phase 3 PERSEUS trial (NCT03710603), which will determine whether subcutaneous Dra in combination with VRd improves PFS (the primary end-point of the trial) as compared with VRd, in TENDMM.

Improving results in the induction phase by incorporating second generation PIs and MoAbs was the hypothesis tested in the MMY1001 trial and in the MASTER study.

MMY1001 EQUULEUS (NCT01998971) was an open-label, phase 1b study, to evaluate the safety, tolerability, and dose of Dara when administered in combination with various treatment regimens in both the ND and RR settings. One of these cohorts evaluated the addition of Dara to carfilzomib, lenalidomide, dexamethasone (KRd) in 22 patients without any evidence of cardiac disease (patients with uncontrolled hypertension, heart failure NYHA III-IV, ischemic cardiac diseases or previous cerebrovascular accident were excluded), irrespective of transplant-eligibility [27]. Dara was administered with a split-first dose and carfilzomib was escalated from 20 mg/mq to 70 mg/mq weekly from Cycle 1 Day 8; Dara-KRd was administered until 13 cycles or elective discontinuation for ASCT, which was performed in six patients. Dara-KRd displayed a high response rate (ORR 100%, 43% sCR), no adverse impact on the harvest and safety profile consistent with the adverse effects already known of Dara and carfilzomib.

The MASTER study (NCT03224507) was a phase 2 study using Dara-KRd in frontline treatment, which adopted an MRD-driven approach and evaluated the impact on outcomes of treatment discontinuation in patients with sustained MRD negativity [28]. The patients received 4 cycles of KRd, single ASCT, no or four or eight cycles KRd of consolidation, according to MRD status. Discontinuation was planned after two consecutive MRD negativity assessments (sustained-MRD), evaluated by NGS, with a sensitivity of 10^−5^ idem as before and monitored for MRD reappearance at 6 and 18 months after discontinuation; otherwise patients still MRD positive after consolidation underwent lenalidomide maintenance. Sixty-nine patients were enrolled; 38 received ASCT and 22 completed the post-transplant assessment. After induction, ≥VGPR was achieved in 92% of patients and CR/sCR in 91% of patients as best response. The MRD negativity rate, evaluated with NGS taking a cut-off of 10-5, was 34%, 70% and 80% after induction, ASCT and at best response, respectively, and 28%, 45%, 65% at the same time-points, taking a cut-off of 10-6. No discontinuation due to adverse events was reported. All 11 patients who achieved confirmed MRD negativity by NGS also attained imaging-MRD negativity as assessed by PET/CT and had neither clinical relapse nor any MRD recurrence, albeit over a short follow-up ranging from 0.8 to 7.3 months. This study is the first report on an MRD response-adapted therapy including MoAbs, showing that sustained MRD negativity may potentially allow one to identify patients who can safely stop maintenance treatment, and hence reduce the number of patients in continuous therapy.

In another phase 2 trial, the combination of Dara with the triplet KRd is currently investigating bi-weekly versus weekly administration of carfilzomib over a total of eight cycles, in NDMM, both transplant eligible and not, with stem cell collection for fit patients after four to six cycles. The primary endpoint is MRD negativity rate detected by MFC and NGS with a sensitivity of 10-5. The study is still on-going; a first analysis after enrollment of the first 29 patients showed that the rate of MRD negativity was 83%, with no differences between the two schedules of administration [29].

The LYRA study (NCT02951819) enrolled 101 patients, including 87 NDMM, either TE or non-TE, as well as 14 relapsed patients, and treated them with Dara in combination with cyclophosphamide, bortezomib and dexamethasone for 8 cycles [30]. Among the TE patients, the VGPR rate after four cycles of induction was 44.2% and increased to 55.8% (9.3% CR) by the end of induction phase, indicating a trend toward a deeper response over time. Median duration of response and median PFS were not reached, with a 12-month PFS and OS rate of 87% and 98.8% months, respectively. The results of this trial suggest that an IMiDs-sparing induction regimen is feasible, with a satisfactory safety profile.

#### 2.5.2. Isatuximab

The anti-CD38 MoAb Isatuximab (Isa) is currently being tested in NDMM by a phase 2 trial in quadruplet combination with KRd, conducted by the German MM Group; the results of the pre-specified interim induction were reported at the 2020 American Society Clinical Oncology (GMMG-CONCEPT trial, NCT03104842) [31]. Only HR MM, defined by the combination of ISS2 or ISS3 clinical stages and deletion 17 or t(4;14) or t(4;16) translocations or >3 copies of 1q21, are enrolled. The protocol consists of six cycles of Isa-KRd induction, four cycles of Isa-KRd consolidation followed by Isa-KR maintenance and is planned to be completed by ASCT in TE patients. The primary end-point is MRD negativity by next generation flow (NGF) at the end of consolidation. The preliminary analysis, including 43 of the planned 153 patients, reported deep responses (46% CR, 44% VGPR) after induction, adequate peripheral blood stem cell collection in the TE group and low rates of non- hematologic toxicities in this hard-to-treat MM population. The final results of the study are eagerly awaited. Table 1 summarizes the main studies available with the use of Dara and Isa in NDTEMM and SMM.

### 2.6. Anti-SLAM F7 Monoclonal Antibodies

Given the proven efficacy of combination regimens including the anti-SLAM F7 MoAb Elo in the setting of RRMM [18,32,33], phase 2 and 3 trials have been designed to explore the incorporation of this agent in first line therapy for NDTEMM patients (Table 2).

A regimen including bortezomib and an IMiD is usually considered the gold standard treatment for patients with NDMM who are fit for high-dose chemotherapy [34], along with post-transplant maintenance therapy employing lenalidomide [35]. The GMMGHD6 phase III trial (NCT02495922) is currently investigating the role of Elo in combination with VRd induction/consolidation and lenalidomide maintenance within a high dose concept [36]. Four treatment strategies are being compared, on the basis of use of Elo in addition to background treatment of VRd induction/consolidation (VRd +/− Elo) as well as lenalidomide maintenance treatment (lenalidomide +/− Elo), regarding PFS as the primary endpoint. The number of patients included is 564; intermediate analysis [37] showed no benefit from the addition of Elo to VRd in terms of response after four cycles of induction therapy. In particular, the ORR and VGPR rates for the VRd plus Elo group as compared with the VRd group were 82% versus 86% and 58% versus 54%, respectively. On the other hand, the addition of Elo did not result in increasing toxicities. The study is nevertheless still going on and results as to MRD, responses at later time points, and PFS are still awaited.

The inclusion of Elo in a transplant setting has also been investigated in a phase 2 trial (NCT02843074) aimed at exploring the feasibility of incorporating Elo-len-dex as induction, consolidation and maintenance therapy into a ASCT program [38]. Elo-len-dex is indeed active, well-tolerated, and approved by FDA and EMA for patients with relapsed myeloma [18]. Fifty-two patients with NDTEMM were enrolled. According to the study design, patients received four cycles of Elo-len-dex induction and four cycles of Elo-len-dex consolidation, before and after ASCT, and then went on to reduced-dose Elo-len-dex maintenance for up to 24 months. The overall toxicity was low, the most common adverse events being fatigue, diarrhea and nausea. The best ORR was 92% (69% ≥ VGPR), and the 18-month estimates for PFS and OS were 83% and 89%, respectively. The best ORR was similar for HR patients -defined as R-ISS III or at least one of del17p, t(4;14), t(14;16)- (87%, including 67% ≥ VGPR), and the standard-risk (93%, including 53% ≥ VGPR) population; however, the median PFS and OS were shorter for the HR group, being 20.5 and 22.0 months, respectively, as compared with median achieved not reached by the standard-group after a median follow-up of 20 months [38].

The role of Elo in the setting of HR disease was specifically addressed in the phase 2 randomized trial SWOG-1211 (NCT01668719). Patients with NDMM and HR disease were enrolled. HR disease was defined by the presence of one or more HR cytogenetic abnormalities (including t(14; 16), t(14; 20), del(17p), and amplification 1q21), and/or poor risk score by gene expression profiling, and/or primary plasma cell leukemia, and/or elevated serum LDH. According to the study design, patients were randomized to receive eight cycles of VRd induction, followed by dose-attenuated VRd maintenance until disease progression, with or without Elo. Stem cell collection was allowed, but ASCT was deferred until progression. After 53 months of follow-up, the primary analysis on 103 evaluable patients [39] showed that the addition of Elo to VRd induction and maintenance did not improve patient outcomes in terms of PFS or OS. The safety profile was similar, with the exception of increased rates of grade ≥3 neutropenia (27% versus 16%), infections (17% versus 8%) and neuropathy (13% versus 8% sensory, and 8% versus 2% motor) in the Elo-VRd arm as compared with VRd. Notably, median PFS (34 months for VRd, and 31 months for VRd plus Elo) and OS (not achieved for VRd and 68 months for VRd plus Elo) exceeded the original statistical assumption, supporting the role of a PI and IMiD combination for HR disease, though Elo seems not to improve the prognosis of this specific population [39].

The administration of Elo-based combination for patients with HR disease was also investigated in a retrospective study that evaluated Elo-based combinations as consolidation treatment after ASCT [40]. Thirty-one patients with HR features—defined by any of ISS or Revised-ISS stage 3, CD-138 selected fluorescence in situ hybridization with del 17p, 1q21 gain, t(4;14), t(14;16), and t(14;20), cytogenetics with 13q del or complex karyotype, and/or HR gene expression profile score -.and who achieved stable disease or better after ASCT, were treated with four cycles of Elo-len-dex (29 patients) or Elo-pomalidomide-dex (two patients), beginning 30-90 days after ASCT. Elo-len-dex and Elo-pomalidomide-dex are approved by FDA and EMA for the treatment of patients with RRMM who have received at least one prior therapy (on the basis of ELOQUENT-2 trial) [18], and at least two prior therapies (ELOQUENT-3 trial) [33], respectively. The hypothesis was that Elo-based consolidation may enhance an immune-competent phenotype, by restoring NK cells and effector T-cell populations at a time of maximal disease de-bulking, and may ultimately improve outcomes among patients with HR markers. Consolidative Elo-lend-dex or Elo-pomalidomide-dex deepened response, compared to post-ASCT, with 71.4% versus 19.4% achieving sCR. MRD negativity was achieved in 19.3% of the patients after consolidation, versus 12.9% of the patients tested before consolidation. With a median follow-up of 24.8 months, median PFS was 31.4 months, similar to or perhaps surpassing historical reports of HR patients receiving lenalidomide maintenance until progression, with the advantage of a short-term fixed duration treatment [40].

Elo-based maintenance therapy following ASCT has also been evaluated in a retrospective study [41]. Seven patients, with ND or RR disease, were treated with Elo-len-dex (five patients), Elo-bortezomib-dex (one patient) or Elo-bortezomib-methylprednisolone (1 patient) following transplant. With a median of 20 cycles, five patients (71.4%) had improvement of depth of response while on therapy, and four patients (57.1%) converted to CR. An Elo-based maintenance regimen improved the rate of a CR or VGPR from 57.1% to 100% before and after maintenance therapy [41]. Given its unique action and rare side effects, further studies incorporating Elo in the post-transplant management of NDMM are warranted.

Additional frontline phase 2 and 3 clinical trials with anti-SLAM F7 MoAb are currently on-going or will start very soon. Elo is currently being evaluated in combination with KRd prior to and following ASCT in a phase 3 trial (NCT03948035). Moreover, a phase 2 trial exploring Elo plus lenalidomide maintenance therapy post ASCT (NCT02420860) is currently ongoing.

## 3. Conclusions

Results from phase 2 and 3 trials clearly show that the addition of MoAbs to already existing standard of care represents an effective and safe therapeutic option not only in RRMM but also in NDTEMM. In particular, the addition of Dara to standard triplets, such as VTd or VRd or investigational KRd, significantly improves depth of response and PFS, without adding significant toxicities or affecting patient quality of life; the results on OS are still immature. The ability of MoAbs to induce higher rates of MRD-negativity is particularly relevant, as this outcome is likely in the near future to be not only a powerful prognostic factor but also a driver of treatment decisions. An outcome improvement has been registered across different patient subgroups, even though with a less pronounced effect in patients bearing an HR cytogenetic profile. In the CASSIOPEIA trial, PFS in patients with HR disease - defined as the presence of del17p and t(4;14)—treated with Dara-VTd and ASCT was inferior to that of patients with standard-risk disease treated with the same regimen [42], although the benefit of Dara-VTd relative to VTd was consistent across pre-specified subgroups both in terms of PFS and MRD negativity rates [10]. In the GRIFFIN trial, the subgroup analysis of MRD negativity rate favored Dara-VRd in all analyzed subgroups but was not statistically significant for ISS stage III disease or HR cytogenetic abnormalities, though the small number of patients in these categories limits the interpretation [26]. Preliminary results of phase 2 study showed deep responses in HR patients treated with Dara-KRd with an MRD-driven approach [28] or Isa-KRd [31]. Anti-CD38 MoAb-based quadruplets are likely to become the new standard of care combined with ASCT in NDTEMM patients. Additional studies are needed to identify the best treatment strategy for HR patients. The clinical results of Elo in the setting of NDTEMM appear still immature. The strong stimulation of NK cell activity exerted by Elo would suggest using this MoAb as a long-term treatment in patients achieving deep response after initial therapy. Elo-combinations are currently investigated as induction, consolidation and maintenance therapy. The results on HR disease are conflicting. Indeed, the addition of Elo to VRd induction and maintenance did not improve patient outcomes in terms of PFS or OS, in the phase 2 SWOG-1211 randomized trial specifically addressed to HR disease [39].

The standard of care for patients with SMM has traditionally been observation until an MDE occurs; however, since two trials demonstrated a significant improvement in survival outcomes by the early use of lenalidomide, several phase 2 and 3 trials are currently on-going with MoAbs, the goal being to delay the progression to active MM and restore the immune system. The preliminary results appear promising, though an adequate follow-up is needed to detect the potential benefit on OS.

## Figures and Tables

**Table 1 pharmaceuticals-13-00451-t001:** Results of selected clinical trials with Daratumumab and Isatuximab for smoldering and newly diagnosed multiple myeloma.

Study	Phase	N pts	Design	Patients	Response	Survival Outcomes
**SMM**						
CENTAURUS trial NCT02316106 Landgren, 2020	2	123	Dara monotherapy in 8-week cycles with three different schedules: long intermediate short	SMM Intermediate or high-risk	CR at 15.8 months: 2.4% vs. 4.9% vs. 0% CR at 25.9 months: 4.9% vs. 9.8% vs. 0%	PFS at 2 years: 89.9% vs. 82.0% vs. 75.3%
**NDMM**						
CASSIOPEIA trial NCT02541383 Moreau, 2019	3	1074	Dara-VTd vs. VTd induction/consolidations plus ASCT	NDTE	sCR: 29% vs. 20% MFC-MRD negativity (10^−5^ cut-off): 64% vs. 44%	PFS at 18 months: 93% vs. 85%
GRIFFIN trial NCT02874742 Voorhees, 2020	2	207	Dara-VRd vs. VRd induction/consolidation plus ASCT—Dara-lena or lena maintenance	NDTE	sCR: 42% vs. 32% NGS-MRD negativity (10^−5^ cut-off): 51% vs. 20%	PFS at 2 years: 95.8% vs. 89.8%
MMY1001 EQUULEUS trial NCT01998971 Chari, 2017	1b	22	Dara-KRd ± ASCT K 20 mg—70 mg/m^2^ 1,8,15	ND TE or non-TE	CR: 57% VGPR: 33%	PFS at 12 months: 95%
MASTER trial NCT03224507 Costa, 2019	2	69	4 Dara-KRd—ASCT—0,4,8 Dara-KRd consolidation according MRD K 56 mg/mq 1,8,15	NDTE	NGS-MRD negativity (10^−5^ cut-off): 34% post-induction, 70% post-ASCT 80% best response NGS-MRD negativity (10^−6^ cut-off): 28% post-induction, 45% post-ASCT 65% best response	Not reported
LYRA trial NCT02951819 Yimer et al., 2019	2	101	Dara-VCD	ND TE or non-TE (87) RR (14)	≥VGPR: 55.8% CR: 9.3%	PFS at 12 months: 87%
GMMG-CONCEPT trial, NCT03104842 Weisel, 2020	2	152 (50 evaluable)	6 Isa-KRd +/− ASCT + 4 Isa-KRd + Isa-KR maintenance	ND TE or non-TE HR	VGPR: 44% CR: 46%	Not reported

N pts: number of patients; SMM: smoldering multiple myeloma; Dara: Daratumumab; CR: complete response; PFS: progression-free survival; NDMM: newly diagnosed multiple myeloma; Dara-VTd: Daratumumab-bortezomib-thalidomide-dexamethasone; VTd: bortezomib-thalidomide-dexamethasone; ASCT: autologous stem cell transplantation; NDTE: newly diagnosed transplant eligible; sCR: stringent complete response; MFC: multiparametric flow cytometry; MRD: minimal residual disease; Dara-VRd: Daratumumab-bortezomib-lenalidomide-dexamethasone; VRd: bortezomib-lenalidomide-dexamethasone; Dara-lena: daratumumab-lenalidomide; Lena: lenalidomide; NGS: next generation sequencing; Dara-KRd: Daratumumab-carfilzomib-lenalidomide-dexamethasone; K: carfilzomib; VGPR: very good partial response; Dara-VCD: Daratumumab-bortezomib-cyclophosphamide-dexamethasone; RR: relapsed/refractory; Isa-KRd: Isatuximab-carfilzomib-lenalidomide-dexamethasone; Isa-KR: Isatuximab-lenalidomide; HR: high-risk.

**Table 2 pharmaceuticals-13-00451-t002:** Results of selected clinical trials with elotuzumab for smoldering and newly diagnosed multiple myeloma.

Study	Phase	N pts	Design	Patients	Response	Survival Outcomes
**SMM**						
NCT02960555 Jagannath, 2018	2	31	Elo monotherapy 20 mg/kg (days 1 and 8 cycle 1, monthly from cycle 2) 10 mg/kg (weekly cycles 1 and 2, twice monthly from cycle 3)	SMM HR	ORR: 10%	PFS at 2 years: 69%
NCT02279394 Liu, 2018	2	50	Elo-len vs. Elo-len-dex	SMM HR	ORR: 84% VGPR: 37% CR: 6%	None of the patients progressed to MM at 3 years
**NDMM**						
GMMGHD6 trial NCT02495922 Goldschmidt, 2020	3	564	VRd induction/consolidation (VRd +/− Elo), ASCT, lenalidomide maintenance treatment (lenalidomide +/− Elo)	NDTE	Elo plus VRd vs. VRd induction ORR: 82% vs. 86% VGPR: 58% vs. 54%	Not reported
NCT02843074 Berdeja, 2019	2	52	Elo-len-dex plus ASCT	NDTE	ORR: 92% ≥VGPR: 69% HR: ORR 87%, SR: ORR 93%	PFS at 18 months: 83% OS at 18 months: 89% HR patients median PFS: 20.5 months median OS: 22 months SR patients: median PFS and OS not reached with 20 months of follow-up
SWOG-1211 trial NCT01668719 Usmani, 2020	2	103	Elo plus VRd vs. VRd	ND TE or non-TE HR	Elo plus VRd vs. VRd ORR: 83% vs. 88% ≥VGPR: 23% vs. 26%	Elo plus VRd vs. VRd median PFS: 31 vs. 34 months median OS: 68 months vs. not reached

N pts: number of patients; SMM: smoldering multiple myeloma; Elo: Elotuzumab; HR: high-risk; ORR: overall response rate; PFS: progression-free survival; Elo-len: Elotuzumab-lenalidomide; Elo-len-dex: Elotuzumab-lenalidomide-dexamethasone; ASCT: autologous stem cell transplantation; CR: complete response; VGPR: very good partial response; NDMM: newly diagnosed multiple myeloma; VRd: bortezomib-lenalidomide-dexamethasone; NDTE: newly diagnosed transplant eligible; SR: standard-risk; OS: overall survival.

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
