# Peer review of "The Role of Monoclonal Antibodies in Smoldering and Newly Diagnosed Transplant-Eligible Multiple Myeloma"

_pharmaceuticals, 2020, doi:10.3390/ph13120451_

Round 1

Reviewer 1 Report

Well-written comprehensive and timely review of the current state of the use of monoclonal antibodies in multiple myeloma, with specific focus on smoldering myeloma and newly diagnosed transplant-eligible patients. Only a few very minor comments: Line 24: "Majo" should be corrected to "Mayo"; line 28: "failed to any show" should be corrected to "failed to show any". Otherwise, looks good.

Author Response

We would like to thank the Reviewer for the positive feedback. Text was modified accordingly.

Reviewer 2 Report

In general, this is a well-written, comprehensive review of the use of monoclonal antibodies in smoldering myeloma and newly diagnosed transplant-eligible myeloma.

A few comments:

  1. Section 2.3 should include the fact that the combination of elotuzumab/pomalidomide/dexamethasone is approved for relapsed/refractory myeloma with at least 2 prior lines of therapy based on the ELOQUENT-3 trial.
  2. Discussion of the GRIFFIN study. The authors should clarify what the primary endpoint of this study was. The confidence intervals for the 24 month PFS rate estimates should be included. As currently stated, the authors imply that the Dara-VRD arm is superior to the VRD arm from a PFS perspective, but there is not a statistically significant difference. The authors should describe the design of the PERSEUS study is and indicate what its primary endpoint is.
  3. The EQUULEUS trial was a multi-cohort study evaluating different daratumumab-based combinations in both the newly diagnosed and relapsed/refractory settings. The authors should specify that one of these cohorts evaluated DaraKRD.
  4. In the discussion of SWOG-1211. The authors should include discussion of the increased rates of > grade 3 neutropenia and infections observed in the Elo-RVD arm.
  5. In the summary, would recommend more discussion about the lack of benefit from the addition of these mAbs to patients with high-risk cytogenetics.

Author Response

We would like to thank the Reviewer for the positive feedback and suggestions. All the comments have been addressed.

    Section 2.3 should include the fact that the combination of elotuzumab/pomalidomide/dexamethasone is approved for relapsed/refractory myeloma with at least 2 prior lines of therapy based on the ELOQUENT-3 trial.

According to Reviewer’s suggestion, the following sentence has been added: Elo-len-dex and Elo-pomalidomide-dex are approved by FDA and EMA for the treatment of patients with RRMM who have received at least 1 prior therapy (on the basis of ELOQUENT-2 trial) [18], and at least 2 prior therapies (ELOQUENT-3 trial) [33], respectively. “

    Discussion of the GRIFFIN study. The authors should clarify what the primary endpoint of this study was. The confidence intervals for the 24 month PFS rate estimates should be included. As currently stated, the authors imply that the Dara-VRD arm is superior to the VRD arm from a PFS perspective, but there is not a statistically significant difference. The authors should describe the design of the PERSEUS study is and indicate what its primary endpoint is.

We thank the Reviewer for these suggestions. Text was modified accordingly: “The primary end-point, sCR rate by the end of post-ASCT consolidation, favored Dara-VRd versus VRd (42.4% versus 32.0%; odds ratio 1.57; 95% CI 0.87-2.82; 1-sided p=0.068) …… Median PFS was not reached, with 24-month PFS estimates of 95.8% (95% CI, 89.2-98.4) for the Dara-VRd group and 89.8% (95% CI, 77.1-95.7) for the VRd group. The positive results from GRIFFIN study support the ongoing phase 3 PERSEUS trial (NCT03710603), which will determine whether subcutaneous Dra in combination with VRd improves PFS (the primary end-point of the trial) as compared with VRd, in TENDMM.”

    The EQUULEUS trial was a multi-cohort study evaluating different daratumumab-based combinations in both the newly diagnosed and relapsed/refractory settings. The authors should specify that one of these cohorts evaluated DaraKRD.

According to Reviewer’s suggestion, the following sentence has been added: “MMY1001 EQUULEUS (NCT01998971) was an open-label, phase 1b study, to evaluate the safety, tolerability, and dose of Dara when administered in combination with various treatment regimens in both the ND and RR settings. One of these cohorts…”

    In the discussion of SWOG-1211. The authors should include discussion of the increased rates of > grade 3 neutropenia and infections observed in the Elo-RVD arm.

As suggested, we added in the text: “The safety profile was similar, with the exception of increased rates of grade ≥3 neutropenia (27% versus 16%), infections (17% versus 8%) and neuropathy (13% versus 8% sensory, and 8% versus 2% motor) in the Elo-VRd arm as compared with VRd.”

    In the summary, would recommend more discussion about the lack of benefit from the addition of these mAbs to patients with high-risk cytogenetics.

We thank the Reviewer for this suggestion. A more detailed discussion on HR disease has been added in the “conclusion” section.